# Comparison of 8- versus 12-weeks of glecaprevir/pibrentasvir for Taiwanese patients with hepatitis C and compensated cirrhosis in a real-world setting

**Yung-Hsin Lu**[1], **Chung-Kuang Lu**[1], **Chun-Hsien Chen**[1], **Yung-Yu Hsieh**[1], **Shui-Yi Tung**[1,2], **Yi-Hsing Chen**[1], **Chih-Wei Yen**[1], **Wei-Lin Tung**[1], **Kao-Chi Chang**[1], **Wei-Ming Chen**[1], **Sheng-Nan Lu**[1,2], **Chao-Hung Hung**[1,2], **Te-Sheng Chang**[1,2]*

1 Department of Internal Medicine, Division of Gastroenterology and Hepatology, Chang Gung Memorial Hospital, Chiayi, Taiwan, 2 College of Medicine, Chang Gung University, Taoyuan, Taiwan

* cgmh3621@cgmh.org.tw

**Data Availability Statement:** All relevant data are within the paper and its Supporting Information files.

## Abstract

Real-world data on the effectiveness of glecaprevir/pibrentasvir (GLE/PIB) for patients with HCV infection and compensated cirrhosis is limited, especially for the 8-week regimen and in an Asian population. This retrospective study enrolled 159 consecutive patients with HCV and compensated cirrhosis who were treated with GLE/PIB at a single center in Taiwan. Sustained virological response (SVR) and adverse events (AEs) were evaluated. Among the 159 patients, 91 and 68 were treated with GLE/PIB for 8 and 12 weeks, respectively. In the per protocol analysis, both the 8- and 12-week groups achieved 100% SVR (87/87 vs. 64/64); and in the evaluable population analysis, 95.6% (87/91) of the 8-week group and 94.1% (64/68) of the 12-week group achieved SVR. The most commonly reported AEs, which included pruritus (15.4% vs. 26.5%), abdominal discomfort (9.9% vs. 5.9%), and skin rash (5.5% vs. 5.9%), were mild for the 8- and 12-week groups. Two patients in the 8-week group exhibited total bilirubin elevation over three times the upper normal limit. One of these two patients discontinued GLE/PIB treatment after 2 weeks but still achieved SVR. Both 8- and 12-week GLE/PIB treatments are safe and effective for patients of Taiwanese ethnicity with HCV and compensated cirrhosis.

## Introduction

Chronic hepatitis C virus (HCV) infection is a major public health problem affecting approximately 110 million individuals worldwide, with an estimated 71.1 million of them living with active viremic infection in 2015 [1]. HCV is not only a leading cause of liver cirrhosis and hepatocellular carcinoma (HCC) but also reduces quality of life and increases the mortality rates of many hepatic and extrahepatic diseases [2–4]. In the absence of an effective vaccine, attaining a sustained virological response (SVR) is the only means of eliminating HCV-related complications [4]. Before 2014, HCV treatment centered on the use of interferon-based

**Funding:** This work was supported by research grant CORPG6L0101 from Chang Gung Memorial Hospital to TSC.The funder plays no role in the study design, data collection and analysis, decision to publish, or preparation of the manuscript.

regimens, but these have suboptimal efficacies, long treatment durations, and substantial toxicities. Since then, the advent of direct-acting antiviral (DAA) therapy has changed the landscape of HCV treatment such that an SVR is now defined by an undetectable serum HCV ribonucleic acid (RNA) level 12 weeks after completing the DAA regimen [5].

The emergence of all-oral DAAs in anti-HCV therapeutics with SVR rates reaching 95%–99% across all HCV genotypes has presented the prospect of eliminating HCV as a public health threat [6]. The introduction of pangenotypic regimens, characterized by simplified anti-HCV therapy and even higher SVR rates regardless of the genotype, offers new opportunities for the public health response to HCV infection [7, 8]. Glecaprevir (GLE) is a NS3/4A protease inhibitor that targets the viral RNA replication. Pibrentasvir (PIB) is a novel next-generation NS5A inhibitor with potent pangenotypic activity. The fixed-dose combination of GLE/PIB (100/40 mg, Maviret, Fournier Laboratories Ireland Limited, Anngrove, Carrigtwohill, Cork, Ireland) is a once-daily, ribavirin-free, revolutionary pangenotypic DAA regimen approved for a duration of 8, 12, or 16 weeks for patients with HCV [9].

Shortened DAA treatment duration has been reported to reduce the burden on health care resources, overcome elimination barriers, and allow more patients to be treated [10, 11]. GLE/PIB is the only DAA regimen that has been widely accepted for a short treatment duration of 8 weeks for patients with HCV of all genotypes and without cirrhosis [11–13]. On the basis of the high SVR rate of 99.7% obtained in the phase 3 EXPEDITION-8 trial, an 8-week GLE/PIB regimen for treatment-naïve patients with HCV and compensated cirrhosis was approved by the US Food and Drug Administration (FDA) in September 2019 and by the Taiwan FDA (TFDA) in April 2020 [14]. However, real-world data regarding the effectiveness and safety of the 8-week GLE/PIB regimen in patients with HCV and compensated cirrhosis is highly limited, especially in the Asian population. In the present study, we assessed the effectiveness and safety of GLE/PIB for a treatment duration of either 8 or 12 weeks in Taiwanese patients with HCV and compensated cirrhosis in a real-world setting.

## Materials and methods

### Patients and management

As part of an effort to eliminate HCV by 2025 in Taiwan, a nationwide government-funded program was initiated in 2017 by the Taiwan National Health Insurance (NHI) Administration for treating chronic HCV infection with DAAs [15]. In the program's first 2 years, only patients with chronic HCV infection and advanced fibrotic or cirrhotic liver disease were enrolled. The program was extended to all patients with HCV and active viremic infection in 2019, regardless of the duration or severity of liver disease. The only exclusion criterion was for those with a life expectancy of < 6 months. GLE/PIB has been reimbursed by the Taiwan NHI program since August 1, 2018, with prescriptions of 12 weeks for patients with genotypes 1, 2, and 4–6 with compensated cirrhosis and 8 weeks for all patients without cirrhosis. A 16-week treatment was approved for patients with genotype 3 who were previously exposed to treatment with pegylated interferon plus ribavirin. The treatment duration was changed to 8 weeks for genotype 1, 2, and 4–6 treatment-naïve patients on April 1, 2020, and for treatment-naïve patients of all genotypes without history of liver decompensation on August 1, 2020. In this retrospective cohort study, we enrolled patients with HCV and compensated cirrhosis who were aged ≥ 20 years and received GLE/PIB treatment from August 2018 through January 2021 at Chang Gung Memorial Hospital, Chiayi, Taiwan. This study was approved by the Institutional Review Board of Chang Gung Medical Foundation (approval no.: 202100808B0) and was conducted in accordance with the principles of the Declaration of Helsinki and the International Conference on Harmonization for Good Clinical Practice.

Baseline patient demographic data and on-treatment information, including laboratory parameters and adverse events (AEs), were obtained from the hospital's electronic medical records. Treatment for patients with HCV of various genotypes with a fixed-dose combination of GLE/PIB was determined by the treating physician on the basis of the labels approved by the TFDA, which were in compliance with the standard of care stipulated by international guidelines on HCV infection [11]. In brief, GLE/PIB was prescribed for a duration of either 8 or 12 weeks according to the patient's cirrhosis status and treatment experience with interferon-based regimens. For this study, the cirrhosis diagnosis was determined using either a fibrosis-4 index (FIB-4) score of $> 6$ or a FIB-4 score of $> 3.25$ with at least two pieces of clinical, radiological, endoscopic, or laboratory evidence of cirrhosis or portal hypertension. Patients with Child–Pugh B or C cirrhosis classification or a history of liver decompensation were contraindicated for GLE/PIB and not included for treatment. All patients gave written informed consent prior to the initiation of DAA therapy.

## Outcome evaluation

The primary endpoint was the rate of SVR, which was defined as the proportion of patients with undetectable serum HCV RNA levels 12 weeks after treatment cessation, as determined by per protocol (PP) analysis (participants who received $\geq 1$ dose of DAA with HCV RNA data at posttreatment week 12) or evaluable population (EP) analysis (participants who received $\geq 1$ dose of DAA with at least one available postbaseline response assessment). Biochemical response was defined as proportion of patients with normalization of alanine aminotransferase (ALT) or aspartate aminotransferase (AST) by PP analysis. The estimated glomerular filtration rate (eGFR) was assessed by Modification of Diet in Renal Disease (MDRD) Study equation for all eligible patients at baseline, the end of treatment (EOT), and off-treatment weeks 12 for both the 8-week and 12-week groups. The incidence of AEs of all evaluable population and the proportion of undetectable serum HCV RNA levels at the EOT were also investigated.

## Statistical analysis

Continuous variables were expressed as medians and interquartile ranges. Descriptive characteristics were expressed as numbers (percentages) for the categorical variables. Differences between groups were analyzed using the chi-square test or Fisher's exact test for categorical variables and the Mann–Whitney $U$ test for continuous variables. Wilcoxon rank sum test was used to compare the dynamic changes of eGFR between the 8- and 12-week groups. Statistical analyses were performed using SPSS software (version 22.0; IBM, Chicago, IL, USA). The $p$-value cutoff for statistical significance was defined as $<0.05$.

## Results

### Patient baseline characteristics

As depicted in Fig 1, a total of 1015 patients received GLE/PIB treatment at Chang Gung Memorial Hospital (Chiayi) from August 2018 through January 2021. Among them, 889 patients received the 8-week treatment and 126 patients received the 12-week treatment. This study enrolled all 159 patients with HCV and compensated cirrhosis; 91 and 68 patients received the 8- and 12-week treatments, respectively. Table 1 displays the baseline characteristics of these 159 patients. Compared with the group treated with the 12-week regimen ($n = 68$), the group treated with the 8-week regimen ($n = 91$) included more treatment-naïve patients (87/91 or 95.6% vs. 57/68 or 83.8%, $p = 0.03$), fewer patients with previous hepatocellular

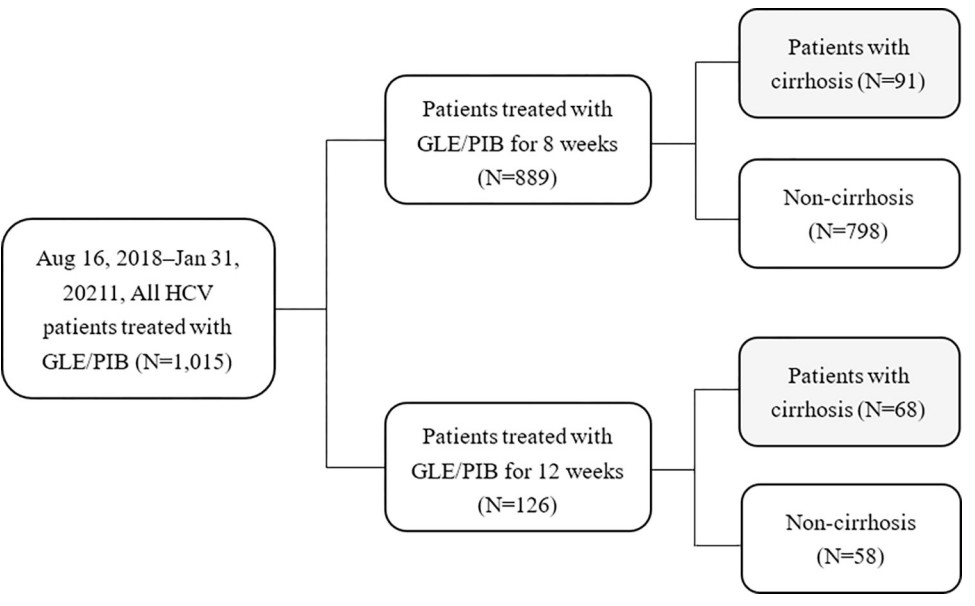

**Fig 1. Study flow diagram.**

carcinoma (HCC) (5/91 or 5.5% vs. 16/68 or 23.5%, $p = 0.02$), and more patients with Child-Pugh score 5 (83/91 or 91.2% vs. 51/68 or 75%, $p = 0.01$). Moreover, those in the 8-week regimen group had higher hemoglobin (median:13.4, interquartile range:12.2–14.4 vs. 12.4, 11.3–14 g/dL, $p = 0.008$), platelet count (128, 103–142.5 vs. 104.5, 84.8–135 × 10$^9$ cells/L, $p = 0.006$), albumin (4.2, 3.9–4.4 vs. 4.0, 3.7–4.2 g/dL, $p = 0.04$), serum creatinine (0.9, 0.8–1.1 vs. 1.1, 0.9–1.9 mg/dL, $p < 0.001$) and estimated glomerular filtration rate (eGFR, 77.8, 67.2–93.2 vs. 63.8, 33.5–83.4) mL/min/1.73m$^2$, $p < 0.001$). No significant differences were observed between the two groups in terms of age, sex, baseline HCV RNA level, FIB-4 score, white blood cell count, AST, ALT, total bilirubin, alpha-fetoprotein, or rate of HBV coinfection ($p > 0.05$). The distribution of genotypes in both groups were also demonstrated. As shown in Fig 2, the most common HCV genotypes included genotype 2 ($n = 99$, 62.26%), genotype 1b ($n = 40$, 25.16%), and mixed genotype ($n = 13$, 8.18%).

## Effectiveness outcome

Table 2 details the virological and biochemical responses to the 8- and 12-week regimens. The response rates at the EOT were 97.8% (88/90) for the 8-week group and 100% (66/66) for the 12-week group ($p = 0.62$). One 8-week group participant, a 71-year-old treatment-naïve male patient with a FIB-4 score of 4.46, genotype 2 HCV infection, and a baseline HCV RNA of 3,206,099 IU/mL, discontinued treatment on week 3 because of hyperbilirubinemia. His HCV RNA level was 31 IU/mL at EOT and became undetectable 12 weeks after EOT. Another 8-week group member, a 58-year-old treatment-naïve female patient with a FIB-4 score of 5.25, genotype 2 HCV infection, and a baseline HCV RNA of 5,246,990 IU/mL, had detectable but unquantifiable HCV RNA (<15 IU/mL) at EOT and attained SVR. No difference was identified between the 8- and 12-week groups in terms of the SVR rate. According to the EP analysis, the SVR proportions were 95.6% (87/91) for the 8-week group and 94.1% (64/68) for the 12-week group ($p = 0.95$); according to the PP analysis, SVR proportions were 100% for both the 8- and 12-week groups (87/87 vs. 64/46). Eight patients were lost to follow-up, four in each group. In the 8-week group, one patient withdrew from GLE/PIB treatment because of

**Table 1. Baseline patient characteristics.**

| Characteristics | 8 Weeks | 12 Weeks | p value |
|---|---|---|---|
|  | (N = 91) | (N = 68) |  |
| Age, year |  |  |  |
| Median (interquartile range, IQR) | 73 (64–79) | 70.5 (63.8–79) | 0.90 |
| Sex |  |  |  |
| Male/Female | 45/46 | 41/27 | 0.23 |
| Treatment experience (interferon-based) |  |  |  |
| Naïve/Experienced | 87/4 | 57/11 | 0.03 |
| HBV coinfection[a] |  |  |  |
| Absent/Present | 84/7 | 62/6 | 1 |
| Hepatocellular carcinoma history |  |  |  |
| No/Yes | 86/5[b] | 52/16[c] | 0.02 |
| Diabetes mellitus |  |  |  |
| No/Yes | 61/30 | 39/29 | 0.28 |
| HCV RNA, IU/ml |  |  |  |
| < 800,000/ ≥ 800,000 | 32/59 | 25/43 | 0.97 |
| FIB-4, median (IQR) | 4.6 (3.9–6.2) | 5.2 (4.1–6.7) | 0.53 |
| >6/3.25–6[d] | 25/66 | 26/42 | 0.21 |
| Child-Pugh score 5/6 | 83/8 | 51/17 | 0.01 |
| White blood cell count, $10^9$ cells/L, median (IQR) | 4.9 (4.2–6) | 5.1 (3.9–6.7) | 0.70 |
| Hemoglobin, g/dL, median (IQR) | 13.4 (12.2–14.4) | 12.4 (11.3–14) | 0.008 |
| Platelet count, $10^9$ cells/L, median (IQR) | 128 (103–142.5) | 104.5 (84.8–135) | 0.006 |
| Albumin, g/dL, median (IQR) | 4.2 (3.9–4.4) | 4 (3.7–4.2) | 0.004 |
| Total bilirubin, mg/dL, median (IQR) | 0.8 (0.7–1.1) | 1 (0.7–1.2) | 0.12 |
| AST, U/L, median (IQR) | 78 (54–127) | 71 (39.5–96.5) | 0.07 |
| ALT, U/L, median (IQR) | 74 (43.5–121) | 76 (33–113.3) | 0.24 |
| Creatinine, mg/dL, median (IQR) | 0.9 (0.8–1.1) | 1.1 (0.9–1.9) | <0.001 |
| EGFR, mL/min/1.73m², median (IQR) | 77.8 (67.2–93.2) | 63.8 (33.5–83.4) | <0.001 |
| Alpha-fetoprotein, ng/mL, median (IQR) | 5.7 (3.6–8.8) | 5.4 (3–10.4) | 0.87 |
| HCV RNA, IU/mL, median (IQR) | 1,518,700 (347,600–3,620,230) | 1,243,011 (484,063–3,914,436) | 0.74 |
| HCV genotype | Patient number | Patient number | Total |
| 1b | 30 | 10 | 40 (25.16%) |
| 2 | 55 | 44 | 99 (62.26%) |
| 3 | 0 | 4 | 4 (2.52%) |
| 6 | 1 | 2 | 3 (1.89%) |
| Mixed | 5 | 8 | 13 (8.18%) |

[a]: Patients with positive HBsAg.

[b]: Four patients with active hepatocellular carcinoma (HCC), one received transcatheter arterial chemoembolization and three received radiofrequency ablation for HCC during GLE/PIB therapy.

[c]: Four patients with active HCC, none of them received treatment for HCC during GLE/PIB therapy.

[d]: Among the 108 patients with FIB-4 3.25–6, the diagnosis of cirrhosis was made by presence of at least two of the parameters: 1. clinical stigmata, 2. image, 3. esophageal varices, 4. persistent thrombocytopenia. Of them, 17 (15.5%) was diagnosed by 1 + 2, 14 (38%) by 1 + 4, 13 (12%) by 2 + 3, 35 (32.4%) by 2 + 4 and 2 (1.9%) by 3 + 4.

nasal bleeding, one committed suicide at posttreatment week 8, and the other two patients were lost to follow-up for undisclosed reasons. In the 12-week group, one patient withdrew from the GLE/PIB regimen at week 4 because of refusal to submit to blood tests, one died of H1N1 influenza pneumonia at posttreatment week 9, and the remaining two patients were lost

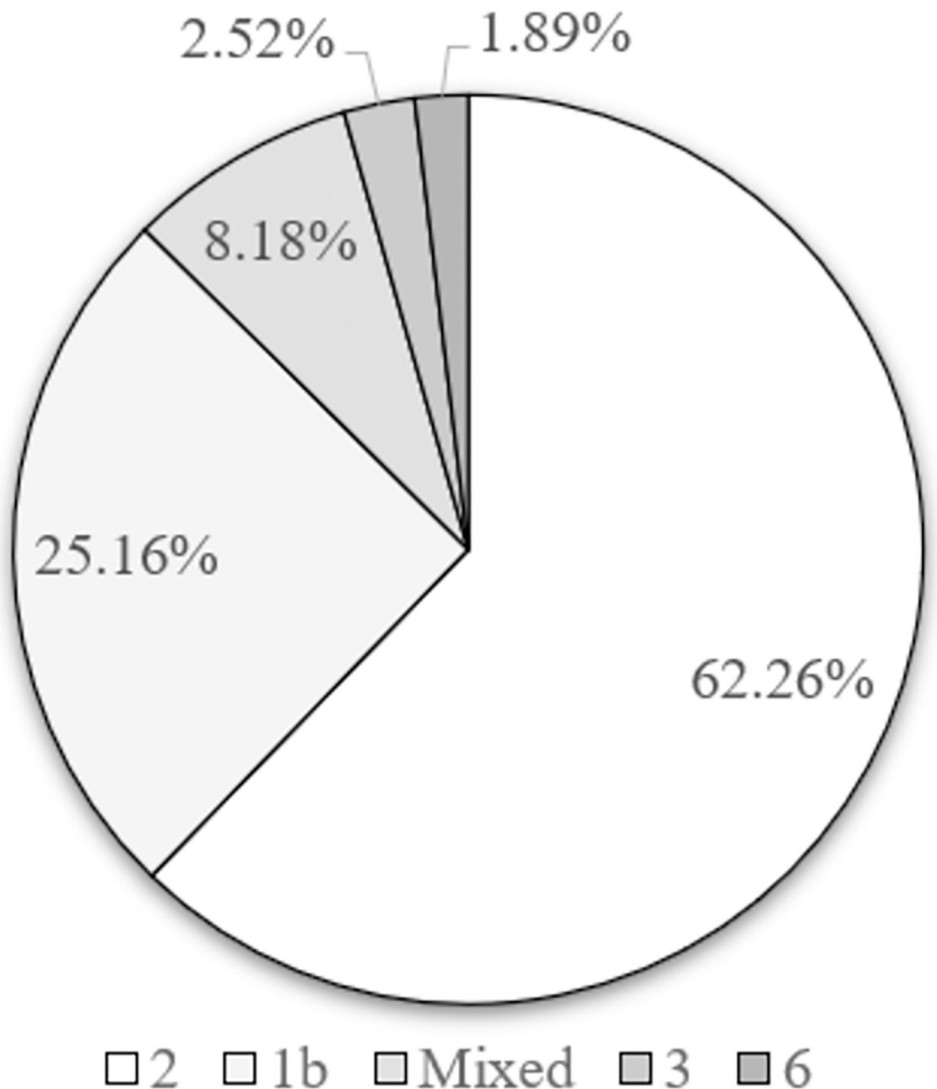

**Fig 2. Genotype distribution.**

to follow-up for undisclosed reasons. The rates of ALT and AST normalization were 82.9% and 73.7% for the 8-week group and 87% and 84% for the 12-week group, respectively.

## Safety outcomes

Table 3 lists the AEs, the most common of which affected > 3% of the patients in the 8-week and 12-week groups and included pruritus (15.4% vs. 26.5%), abdominal discomfort (9.9% vs. 5.9%), skin rash (5.5% vs. 5.9%), fatigue (2.2% vs. 5.9%), and insomnia (2.2% vs. 5.9%). There was no significant difference in the rates of AEs between the two groups of patients ($p > 0.05$). Three patients in the 8-week group and one patient in the 12-week group exhibited over five times the upper normal limit (UNL) elevation of alanine aminotransferase; one patient in the 8-week group also registered over five times the UNL elevation of aspartate aminotransferase. All these AEs had no notable clinical implications. Five patients, three in the 8-week group and two in the 12-week group, had over three times the UNL elevation of direct bilirubin. Clinically significant abnormal laboratory findings were noted in only two patients in the 8-week

**Table 2. Virological and biochemical responses.**

| HCV RNA < LLOD | Patient (N = 91) | Patient (N = 68) | p value |
|---|---|---|---|
| | 8 weeks, n /N (%) | 12 weeks, n /N (%) | |
| **End of Treatment** | | | |
| ETR (PP) | 88/90 (97.8) | 66/66 (100) | 0.62 |
| **12 weeks after Treatment** | | | |
| SVR (EP) | 87/91 (95.6) | 64/68 (94.1) | 0.95 |
| SVR (PP) | 87/87 (100) | 64/64 (100) | |
| Reason for non-SVR, n | | | |
| Relapse | 0 | 0 | |
| Nonresponse | 0 | 0 | |
| Lost to follow-up | 4[a] | 4[b] | |
| End of Treatment[C] | | | |
| ALT normalization (PP) | 58/70 (82.9) | 36/46 (78.3) | 0.7 |
| AST normalization (PP) | 59/76 (77.6) | 39/50 (78) | 1 |
| 12 weeks after Treatment[C] | | | |
| ALT normalization (PP) | 58/70 (82.9) | 40/46 (87) | 0.74 |
| AST normalization (PP) | 56/76 (73.7) | 42/50 (84) | 0.25 |

[a]: One patient withdrew DAA because of nasal bleeding, one committed suicide at posttreatment week 8, and the other two were lost to follow-up for undisclosed reasons.

[b]: One patient withdrew at week 4 because of refusal to submit to blood tests, one died of H1N1 influenza pneumonia at posttreatment week 9, and the other two were lost to follow-up for undisclosed reasons.

[c]: Upper normal limit of AST and ALT was set at 40 IU/mL.

**Table 3. Adverse events.**

| Event, n (%) | 8 weeks (N = 91) | 12 weeks (N = 68) | p value |
|---|---|---|---|
| Adverse events[A] | | | |
| **Pruritus** | 14 (15.4) | 18 (26.5) | 0.13 |
| **abdominal discomfort** | 9 (9.9) | 4 (5.9) | 0.54 |
| **skin rash** | 5 (5.5) | 4 (5.9) | 1 |
| **Fatigue** | 2 (2.2) | 4 (5.9) | 0.4 |
| **Insomnia** | 2 (2.2) | 4 (5.9) | 0.4 |
| **Laboratory adverse event, n (%)** | | | |
| **total bilirubin elevation** | | | |
| **1.5–3 X ULN** | 6 (6.6) | 10 (14.7) | 0.16 |
| **> 3 X ULN** | 2 (2.2) | 0 | 0.61 |
| **direct bilirubin elevation** | | | |
| **1.5–3 X ULN** | 7 (7.7) | 7 (10.3) | 0.77 |
| **> 3 X ULN** | 3 (3.3) | 2 (2.9) | 1 |
| **AST elevation** | | | |
| **3–5 X ULN** | 2 (2.2) | 0 | 0.61 |
| **> 5 X ULN** | 1 (1.1) | 0 | 1 |
| **ALT elevation** | | | |
| **3–5 X ULN** | 1 (1.1) | 0 | 1 |
| **> 5 X ULN** | 3 (3.3) | 1 (1.5) | 0.83 |

[a]: Adverse evets affecting > 3% of the patients.

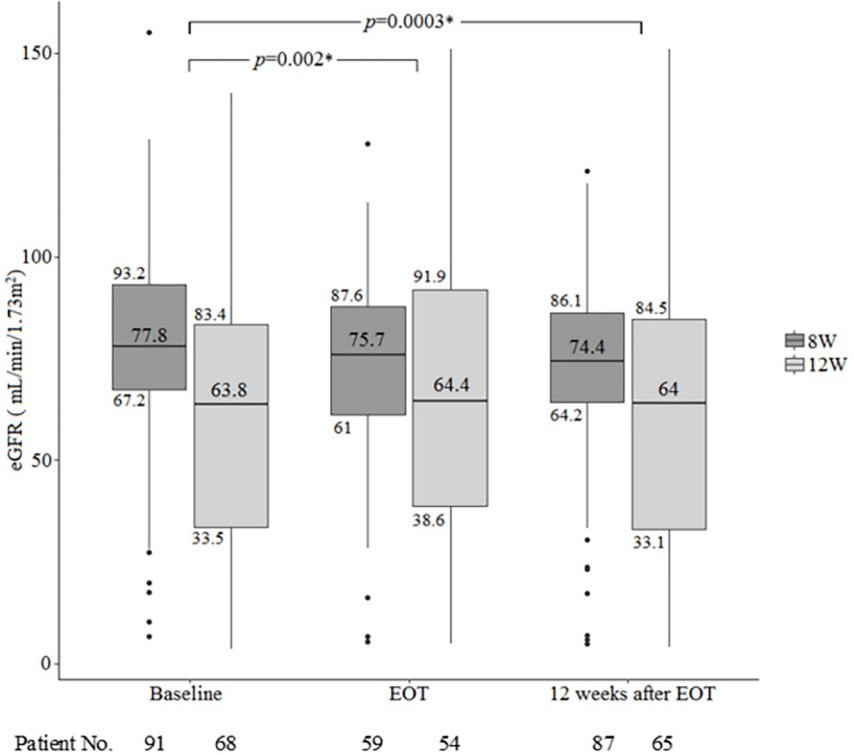

* The 12-week group had a lower dynamic change of eGFR values compared to that of the 8-week group ($p$ = 0.002 for the changes from baseline to EOT and $p$ = 0.0003 for the changes from baseline to 12 weeks after EOT between the two groups).

**Fig 3. Dynamic changes of eGFR during and after GLE/PIB therapy.**

group, who recorded over three times the UNL of serum total bilirubin levels. One of these two patients discontinued GLE/PIB at treatment week 3 with an elevated total bilirubin level of 8.5 mg/dL yet still attained SVR.

## Dynamic changes of eGFR

As shown in Fig 3, the eGFRs (median, interquartile range) at baseline, EOT and 12 weeks after EOT were 77.8 (67.2–93.2, 75.7 (61.0–87.6) and 74.4 (64.2–86.1) ml/min/1.73m$^2$ for the 8-week group and 63.8 (33.5–83.4), 64.4 (38.6–91.9) and 64.0 (33.1–84.5) ml/min/1.73m$^2$ for the 12-week group. The 12-week group had a lower dynamic change of eGFR compared to that of the 8-week group with a $p$ value of 0.002 for the changes between baseline and EOT and a $p$ value of 0.0003 for the changes between baseline and 12 weeks after EOT.

## Discussion

The excellent broad-spectrum antiviral activity, negligible side effects, and simple once-daily dosage of pangenotypic DAA combination regimens simplify health care service delivery and can help accelerate the realization of the HCV elimination goals [16]. Large-scale real-world data from Taiwan also supported the excellent effectiveness and safety profiles of both first-line pangenotypic DAAs, namely GLE/PIB and sofosbuvir/velpatasvir (SOF/VEL) [7, 17–20].

Shortening DAA treatment time regardless of cirrhosis status can further simplify HCV care, which may increase the number of health care professionals who can prescribe DAA regimens and increase the number of patients receiving treatment [10, 11]. The fixed-dose combination of GLE/PIB has demonstrated suitable effectiveness and safety profiles in both clinical trials and real-world settings [7, 14, 17, 19, 21–29]. GLE/PIB is also the only DAA regimen that has been granted NHI reimbursement for an 8-week short-term treatment, applicable to patients with all HCV genotypes and with or without compensated cirrhosis [11–13]. The phase 3 EXPEDITION-8 trial demonstrated that the 8-week GLE/PIB regimen was as effective as the 12-week regimen in treatment-naïve patients with chronic HCV GT1-6 infection and compensated cirrhosis [14]. An integrated analysis of data from eight phase 2 or phase 3 trials revealed that the 8-week regimen of GLE/PIB was efficacious and well tolerated in treatment-naïve patients with HCV genotype 1 to 6 infections, with or without cirrhosis [22]. However, real-world data on the effectiveness of GLE/PIB treatment in patients with HCV and compensated cirrhosis remain limited, especially for the 8-week regimen and in an Asian population [19, 24].

In a retrospective national real-world study from Poland, a modified intent-to-treat analysis for 80 patients with a fibrosis stage of F4 and 954 patients with fibrosis stages of F0–F3 revealed that the 8-week GLE/PIB treatment yielded high SVR rates, comparable to those of the 12-week treatment (98% vs. 96%) [25]. Seven small separate real-world studies from Europe and America, consisting of 135 treatment-naïve patients with HCV and compensated cirrhosis, demonstrated that the 8-week GLE/PIB regimen was associated with a high average SVR rate of 98.1% [26]. In a real-world United State setting, consisting of 71 diverse treatment-naïve, compensated cirrhotic patients, the 8-week GLE/PIB regimen was shown to be safe and highly effective [27]. The real-world data from the German Hepatitis C Registry also indicated that the 8-week GLE/PIB therapy was effective and well-tolerated for HCV patients with compensated cirrhosis [28, 29]. Our results are consistent with these real-world data—both the 8-week and 12-week GLE/PIB treatments were effective (100% SVR for both groups by pp analysis) for patients of Taiwanese ethnicity with HCV and compensated cirrhosis.

Clinical diagnosis of liver cirrhosis is challenging and many noninvasive tools have been developed to reduce the need for liver biopsies. Each of these noninvasive tools has specific advantages and limitations and is considered complementary to the others [30]. In contrast to the aforementioned clinical trials and real-world studies on GLE/PIB, which generally employed the difficult-to-access FibroScan or liver biopsy to assess liver fibrosis, our study used the FIB-4 score of > 6 in combination with clinical, radiological, endoscopic, or laboratory evidence to diagnose cirrhosis. By using the cutoff of 5.36, the FIB-4 score satisfactorily diagnosed liver cirrhosis, namely Metavir fibrosis stage 4, in Asian patients with chronic HCV infection [31]. For patients with FIB-4 scores of 3.25–6 in our study, the diagnosis of cirrhosis was made when at least two out of the following four parameters were identified: clinical (stigmata of chronic liver disease), radiological (signs of portal hypertension on magnetic resonance imaging, computed tomography, or ultrasound), endoscopic (esophageal varices), and laboratory (persistent thrombocytopenia). With these measures, 51 of the 210 patients (24.3%) with a FIB-4 score of > 3.25 were considered to have advanced fibrosis but no definite cirrhosis and were excluded. This result is consistent with the aforementioned study that demonstrated a sensitivity of 72.1% by using 3.8 as the FIB-4 score cut-off in predicting Metavir fibrosis stage 4, suggesting that the cirrhosis diagnosis for the 159 patients enrolled in this study was reliable [31].

In contrast to the overall HCV genotype distribution in Taiwan, wherein genotype 1 is predominant [1], the preponderance of genotype 2 (62.26%) in this study reflected the evolution of DAAs, because the DAA regimens for treating genotype 1 infection were licensed and

reimbursed earlier than those for other genotypes [17, 19]. Although high viral load was previously associated with a high virological failure rate for genotype 2 patients receiving GLE/PIB [32], this phenomenon was not replicated in our genotype 2–dominant cohort, in which PP analysis suggested that all of the patients achieved SVR. The higher number of patients with HCC history and lower levels of hemoglobin, platelet count, and serum albumin in the 12-week group relative to the 8-week group suggest a higher severity of liver disease in this group. This phenomenon might also be explained by the timing of TFDA approval, because the 12-week GLE/PIB regimen was approved earlier than the 8-week regimen and patients who were aware of their liver disease generally had severer disease and were enrolled for treatment as soon as GLE/PIB was approved. This could also explain the lower eGFR in the 12-week group because GLE/PIB was one of the first TFDA-approved DAAs for HCV genotype 2 patients with severe renal impairment [33]. We further analyzed the dynamic changes of eGFR in both groups and found that the 12-week group had a significant lower dynamic change of eGFR compared to that of the 8-week group ($p$ = 0.002 for the changes from baseline to EOT and $p$ = 0.0003 for the changes from baseline to 12 weeks after EOT between the two groups) (Fig 3).

In line with other real-world studies, our study demonstrated that GLE/PIB was well tolerated by patients of Asian ethnicity with cirrhosis [19, 24–29]. A total of four patients did not complete the treatment, and only one of those patients discontinued treatment because of potential GLE/PIB-related hyperbilirubinemia. Despite a short 2-week treatment period, this patient still attained SVR. Other laboratory abnormalities were rare, with approximately 2% of patients with aminotransferases elevation of over five times the UNL and 1.3% of patients with over three times the UNL elevation of total bilirubin level. Consistent with previous studies, pruritus was the most common AE, occurring in 20% of our patients. There was no significant difference in the rates of AEs between the 8-week and 12-week groups.

HBV reactivation during interferon-free DAA treatment for HCV has been a concern for HBsAg-positive patients [34]. We did not provide nucleoside/nucleotide analogue prophylaxis for patients dually infected with HBV and HCV before DAA treatment for HCV in our routine clinical practice. There were 13 patients with positive HBsAg in our study, seven in the 8-week group and six in the 12-week group. Among them, four patients have been undergoing antiviral therapy for HBV before the initiation of GIE/PIB therapy, one in the 8-week group (treated with entecavir) and three in the 12-week group (two with entecavir and one with tenofovir alafenamide). No HBV reactivation was observed during GIE/PIB therapy for these 13 patients.

Finally, since drug-drug interactions (DDIs) have been a significant clinical challenge since the introduction of DAAs, we recorded the concomitant medications in our cohort of 159 patients and evaluated their potential DDIs with currently available new generation pangenotypic DAAs GIE/PIB and SOF/VEL. Overall, red contradictory concomitant medications were met in two patients for the GIE/PIB regimen, one with atorvastatin and the other with lovastatin but none for the SOF/VEL regimen. The result was consistent with the results of a previous research in Taiwan that the potential DDIs between concomitant medications and DAA regimens differed, and sofosbuvir-based regimens had the fewest potential DDIs [35]. Our result indicated that even though the current new generation pangenotypic DAAs have a very high safety profile, careful assessment for potential DDIs is required before and during prescribing the DAAs, especially for the GIE/PIB regimen.

This retrospective real-world study has several limitations. First, selection bias may have occurred because our patients were enrolled from a single referral center. Second, because of inconsistent evaluations and incomplete assessments, the information on AEs was subject to reporting biases, and the causal relationships between the AEs and the treatment could not be fully established. Third, the numbers of some patient subgroups were relatively low because of

the low prevalence rates, particularly those with genotype 3–5 infection. This discrepancy in genotype distribution may limit the generalizability of our study results. Finally, as mentioned earlier, each modality for evaluating liver cirrhosis has certain shortcomings.

## Conclusion

In conclusion, our study demonstrated that both 8- and 12-week GLE/PIB treatments were well tolerated and effective for patients of Asian ethnicity with HCV infection and compensated cirrhosis.

## Supporting information

**S1 Dataset.**
(XLSX)

## Author Contributions

**Conceptualization:** Sheng-Nan Lu, Te-Sheng Chang.

**Data curation:** Chung-Kuang Lu, Chun-Hsien Chen, Yung-Yu Hsieh, Shui-Yi Tung, Yi-Hsing Chen, Chih-Wei Yen, Wei-Lin Tung, Kao-Chi Chang, Wei-Ming Chen, Chao-Hung Hung, Te-Sheng Chang.

**Formal analysis:** Te-Sheng Chang.

**Funding acquisition:** Te-Sheng Chang.

**Methodology:** Chung-Kuang Lu, Chao-Hung Hung, Te-Sheng Chang.

**Supervision:** Te-Sheng Chang.

**Validation:** Te-Sheng Chang.

**Writing – original draft:** Yung-Hsin Lu, Te-Sheng Chang.

**Writing – review & editing:** Te-Sheng Chang.

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
