## [Decision Letter · Decision Letter 0]

27 Jun 2022

PONE-D-22-15455Effectiveness and safety of glecaprevir/pibrentasvir for Taiwanese patients with hepatitis C and compensated cirrhosis in a real-world settingPLOS ONE

Dear Dr. Chang,

Thank you for submitting your manuscript to PLOS ONE. After careful consideration, we feel that it has merit but does not fully meet PLOS ONE’s publication criteria as it currently stands. Therefore, we invite you to submit a revised version of the manuscript that addresses the points raised during the review process.

COMMENTS FROM ACADEMIC EDITOR: The authors should pay more attention in making a more comprehensive review of the published articles, and avoid citing outdated and inadequate references. The content should be significantly reorganized before further consideration. (1) The authors should discuss and compare the current knowledges with regard to first-line pan-genotypic DAAs in Taiwan, including GLE/PIB and SOF/VEL (Liu CH, et al Liver Int 2020, Huang CF et al. Sci Rep 2021; Liu CH, et al Hepatol Int 2021; Cheng PN, et a;. Infect Dis Ther 2022).(2) Delete Reference No. 14 because cirrhosis is not a significant factor to predict SVR in the era of pan-GT DAAs, particularly for HCV GT1 and GT2. (If cirrhosis significantly impact SVR, why GLE/PIB to be shorten from 12 weeks to 8 weeks in compensated cirrhosis? As the authors stated in the study, all cirrhotic patients achieved SVR)(3) Delete Reference No. 18 because it was only a phase II study. Please use Phase III trial GLE/PIB pooled analysis to support your discussion. (Puoti M, et al. J Hepatol 2018;69:293-300)(4) Discussion: Strongly disagree with the illustration "Liver cirrhosis has long been considered a major factor in reducing the SVR rates of HCV treatment even in the era of DAA [14, 19]" Please delete the wordings in the Introduction and Discussion(5) In addition to Reference No., 23, 24. The authors should discuss three published articles (Klinker H, et al. Liver Int 2021;41:1518-22, Flamm SL, et al. Adv Ther 2020;37:2267-74, Zuckerman E, et al. Clin Gastroenterol Hepatol 2020;18:2544-53).(6) Delete Reference No. 27 because it was a poorly written article.  Replace it with two articles (Liu CH, et al Liver Int 2020, Huang CF et al. Sci Rep 2021).(7) Discussion: Strongly disagree with the wording "This could also explain the lower eGFR in the 12-week group because GLE/PIB was one of the first TFDA-approved DAAs for patients with severe renal impairment, especially for those with genotype 2 HCV". Since this study comprised all genotypes, simply attributing to GT2 as the presence of significant difference in eGFR was not correct. The authors should stated a statistical difference for HCV GT2 patients in Results receiving 8 or 12-week GLE/PIB. The original wording should be replaced with "the first TFDA-approved DAAs for HCV GT2 patients with severe renal impairment"(8) How about the potential drug-drug interactions (DDIs) in all these patients? Did these patients take "red contradictory co-medication" before or during GLE/PIB? Please also refer Taiwan DDI data (Liu CH, et al. Aliment Pharmacol Ther 2018) to show a higher risk of DDI compared to SOF-based DAAs, and to alert the healthcare providers to judiciously check DDI before and during GLE/PIB.==============================

We look forward to receiving your revised manuscript.

Kind regards,

Chen-Hua Liu

Academic Editor

PLOS ONE

Journal Requirements:

"This work was supported by research grant CMRPG6J0173 from Chang Gung

Memorial Hospital."

"This work was supported by research grant CMRPG6J0173 from Chang Gung Memorial Hospital to TSC.

The funder plays no role in the study design, data collection and analysis, decision to publish, or preparation of the manuscript"

Reviewers' comments:

Reviewer's Responses to Questions

**Comments to the Author**

1. Is the manuscript technically sound, and do the data support the conclusions?

Reviewer #1: Partly

Reviewer #2: Yes

2. Has the statistical analysis been performed appropriately and rigorously? 

Reviewer #1: No

Reviewer #2: Yes

3. Have the authors made all data underlying the findings in their manuscript fully available?

Reviewer #1: Yes

Reviewer #2: Yes

4. Is the manuscript presented in an intelligible fashion and written in standard English?

Reviewer #1: Yes

Reviewer #2: Yes

5. Review Comments to the Author

Reviewer #1: It is an important mission to eliminate HCV by 2030. Therefore, effectiveness and safety of antiviral therapy are a critical issue. This real-world study pointed that the 8-week GP therapy were effectiveness as the 12-week GP therpay in patients with compensated liver cirrhosis. However, the novelty of this observative and descriptive study is relative low. There are some weak points to overcome.

[major problems]

1. The patients treated with the 8-week GP had higher platelet count, albumin level, Hb and eGFR values; lower proportion of prior HCC. It seemed that the patients receiving 8-week GP had less severe liver fibrosis/cirrhosis.

a). Please explain the reason(s)

b). List Child-Pugh score (Table 1)

c). Provide additional information (such as liver stiffness measurement or Fibrotest)

2. The clinical data is not normal distribution. Please use median (Q1-Q3; Table 1).

3. Please list the total and subgroup numbers of patients with FIB-4 >6 and FIB-4 between 3.25 and 6; please list detail numbers and percentage of the parameters for patients with a FIB-4 3.5-6.

4. The difference of each AE between the 8- and 12-week GP therapy should be compared with P-value. (Table 3)

5. Any predictors of each/total AE? The predictors of clinical significant AE (such as ALT > 5 xULN) and direct hyperbilirubinemia (> 1.5 xUNL) should be investigated. (additional information is necessary)

[minor problems]

1. It should be effectiveness "and safety" of GP for a treatment in patients with HCV and compensated cirrhosis. (line 12 & 14, page 4)

2. What dose superscript "6" mean? (line 3, page 5)

3. Please give references to support "...this AE (pruritis) can increase with longer treatment duration (line 7, page

12)."

4. Indirect hyperbilirubinemia is less clinically significant. Please list total bilirubin > 1.5 x or 3 x ULN AND direct hyperbilirubinemia. (Table 3)

5. Footnotes of Table 1 and 3 were missed.

6. The dataset in the Supporting Information was Chinese records.

Reviewer #2: This study aimed to compare the treatment efficacy and safety between 8-week and 12-week GLE/PIB therapy for patients with compensated HCV-related liver cirrhosis. The authors showed that both 8-week and 12-week GLE/PIB therapy provide excellent antiviral efficacy and safety profiles. Although it is interesting and has clinical implications, several concerns need to be clarified.

1. In this study, 7 patients had HBV and HCV dual infections. Were they treated with antiviral therapy for HBV? How many patients developed HBV reactivation during GIE/PIB therapy for HCV infection?

2. Moreover, 5 patients had a history of HCC. Were they active or non-active during GLE/PIB therapy? Did they undergo therapy for HCC during DAA therapy?

3. How many patients had clinically significant portal hypertension (CSPH)? Were the rates of CSPH different between these two groups of patients?

4. Were the rates of biochemical response or ALT normalization different between these two groups of patients?

5. In this study, patients in the 12-week treatment group had a lower baseline eGFR than those in the 8-week treatment group. Were the dynamic changes of eGFR during and after DAA therapy different between these two groups of patients?

6. It is suggested to provide a table to compare the laboratory data (such as albumin, bilirubin, ALT, platelet count, FIB-4, eGFR, etc.) at the time of end of follow-up (SVR12) between 8-week and 12-week treatment groups.

7. In Table 1, it is suggested to provide the exact p value instead of p< 0.05.

6. PLOS authors have the option to publish the peer review history of their article (what does this mean?). If published, this will include your full peer review and any attached files.

Reviewer #1: No

Reviewer #2: No

---

## [Author Response · Author response to Decision Letter 0]

14 Jul 2022

Response to Reviewers (PONE-D-22-1545) R1

Comparison of 8- versus 12-weeks of glecaprevir/pibrentasvir for Taiwanese patients with hepatitis C and compensated cirrhosis in a real-world setting

Yung-Hsin Lu1, Chung-Kuang Lu1, Chun-Hsien Chen1, Yung-Yu Hsieh1, Shui-Yi Tung1,2, Yi-Hsing Chen1, Chih-Wei Yen1, Wei-Lin Tung1, Kao-Chi Chang1, Wei-Ming Chen1, Sheng-Nan Lu1,2,, Chao-Hung Hung1,2, Te-Sheng Chang1,2*

1Division of Gastroenterology and Hepatology, Department of Internal Medicine, Chang Gung Memorial Hospital, Chiayi, Taiwan

2College of Medicine, Chang Gung University, Taoyuan, Taiwan

Corresponding author: Te-Sheng Chang, MD, PhD, No. 6, Section West, Chiapu Road, Puzi, Chiayi, 613, Taiwan, Tel: 886-5-3621000 Ext. 2005, Email: cgmh3621@cgmh.org.tw

Dear Dr. Chang,

Thank you for submitting your manuscript to PLOS ONE. After careful consideration, we feel that it has merit but does not fully meet PLOS ONE’s publication criteria as it currently stands. Therefore, we invite you to submit a revised version of the manuscript that addresses the points raised during the review process.

COMMENTS FROM ACADEMIC EDITOR: The authors should pay more attention in making a more comprehensive review of the published articles, and avoid citing outdated and inadequate references. The content should be significantly reorganized before further consideration.

(1) The authors should discuss and compare the current knowledges with regard to first-line pan-genotypic DAAs in Taiwan, including GLE/PIB and SOF/VEL (Liu CH, et al Liver Int 2020, Huang CF et al. Sci Rep 2021; Liu CH, et al Hepatol Int 2021; Cheng PN, et a;. Infect Dis Ther 2022).

Answer: Discussion regarding the real-world experiences of the first-line pangenotypic DAAs, GLE/PIB and SOF/VEL, in Taiwan is added in the “Discussion”. (P12L6-8)

(2) Delete Reference No. 14 because cirrhosis is not a significant factor to predict SVR in the era of pan-GT DAAs, particularly for HCV GT1 and GT2. (If cirrhosis significantly impact SVR, why GLE/PIB to be shorten from 12 weeks to 8 weeks in compensated cirrhosis? As the authors stated in the study, all cirrhotic patients achieved SVR)

Answer: The authors agree that growing evidence has proven that cirrhosis is not a significant issue any more in predicting SVR for current new-generation pangenotypic DAAs. We have delete reference No. 14 as the editor suggested. 

(3) Delete Reference No. 18 because it was only a phase II study. Please use Phase III trial GLE/PIB pooled analysis to support your discussion. (Puoti M, et al. J Hepatol 2018;69:293-300)

Answer: Thank you for kind reminder. The previous reference No. 18 is replaced by the updated phase III trial report (Puoti M, et al. J Hepatol 2018;69:293-300).

(4) Discussion: Strongly disagree with the illustration "Liver cirrhosis has long been considered a major factor in reducing the SVR rates of HCV treatment even in the era of DAA [14, 19]" Please delete the wordings in the Introduction and Discussion

Answer: The wordings specified by the editor in the “Introduction” and “Discussion” are deleted.

(5) In addition to Reference No., 23, 24. The authors should discuss three published articles (Klinker H, et al. Liver Int 2021;41:1518-22, Flamm SL, et al. Adv Ther 2020;37:2267-74, Zuckerman E, et al. Clin Gastroenterol Hepatol 2020;18:2544-53).

Answer: The three published articles specified by the editor are addressed and discussed in the “Discussion”. (P13L10-15, P13L22-25)

(6) Delete Reference No. 27 because it was a poorly written article. Replace it with two articles (Liu CH, et al Liver Int 2020, Huang CF et al. Sci Rep 2021).

Answer: The previous reference No. 27 is replaced with the two articles specified by the editor.

(7) Discussion: Strongly disagree with the wording "This could also explain the lower eGFR in the 12-week group because GLE/PIB was one of the first TFDA-approved DAAs for patients with severe renal impairment, especially for those with genotype 2 HCV". Since this study comprised all genotypes, simply attributing to GT2 as the presence of significant difference in eGFR was not correct. The authors should stated a statistical difference for HCV GT2 patients in Results receiving 8 or 12-week GLE/PIB. The original wording should be replaced with "the first TFDA-approved DAAs for HCV GT2 patients with severe renal impairment"

Answer: The original sentence has been changed to the precise wording assigned by the editor.

(8) How about the potential drug-drug interactions (DDIs) in all these patients? Did these patients take "red contradictory co-medication" before or during GLE/PIB? Please also refer Taiwan DDI data (Liu CH, et al. Aliment Pharmacol Ther 2018) to show a higher risk of DDI compared to SOF-based DAAs, and to alert the healthcare providers to judiciously check DDI before and during GLE/PIB.

Answer: The importance regarding careful assessment for potential DDIs before prescribing the DAAs, especially for the GIE/PIB regimen is addressed in the “Discussion”. (P16L6-16)

We look forward to receiving your revised manuscript.

Kind regards,

Chen-Hua Liu

Academic Editor

PLOS ONE

Journal Requirements:

Answer: The format of the manuscript has been modified to meet PLOS ONE’s style requirements.

"This work was supported by research grant CMRPG6J0173 from Chang Gung

Memorial Hospital."

"This work was supported by research grant CMRPG6J0173 from Chang Gung Memorial Hospital to TSC.

The funder plays no role in the study design, data collection and analysis, decision to publish, or preparation of the manuscript"

Answer: The funding statement is deleted and all amended statements are included in the cover letter. 

Reviewers' comments:

Reviewer's Responses to Questions

Comments to the Author

1. Is the manuscript technically sound, and do the data support the conclusions?

Reviewer #1: Partly

Reviewer #2: Yes

2. Has the statistical analysis been performed appropriately and rigorously?

Reviewer #1: No

Reviewer #2: Yes

3. Have the authors made all data underlying the findings in their manuscript fully available?

Reviewer #1: Yes

Reviewer #2: Yes

4. Is the manuscript presented in an intelligible fashion and written in standard English?

Reviewer #1: Yes

Reviewer #2: Yes

5. Review Comments to the Author

Reviewer #1: It is an important mission to eliminate HCV by 2030. Therefore, effectiveness and safety of antiviral therapy are a critical issue. This real-world study pointed that the 8-week GP therapy were effectiveness as the 12-week GP therpay in patients with compensated liver cirrhosis. However, the novelty of this observative and descriptive study is relative low. There are some weak points to overcome.

[major problems]

1. The patients treated with the 8-week GP had higher platelet count, albumin level, Hb and eGFR values; lower proportion of prior HCC. It seemed that the patients receiving 8-week GP had less severe liver fibrosis/cirrhosis.

a). Please explain the reason(s)

Answer: We did explain the reasons in the “Discussion”: This phenomenon might also be explained by the timing of TFDA approval, because the 12-week GLE/PIB regimen was approved earlier than the 8-week regimen and patients who were aware of their liver disease generally had severer disease and were enrolled for treatment as soon as GLE/PIB was approved.(P15L3-9)

b). List Child-Pugh score (Table 1)

Answer: Child-Pugh score was added to Table 1 as the reviewer suggested.

c). Provide additional information (such as liver stiffness measurement or Fibrotest)

Answer: For the limitation of facility in routine clinical practice, Fibroscan and serum markers for the generation of Fibrotest including α2-macroglobulin, haptoglobin, apolipoprotein A1, and gamma glutamyl transpeptidase are not available. 

2. The clinical data is not normal distribution. Please use median (Q1-Q3; Table 1).

Answer: The clinical data is expressed as median (interquartile range) instead of mean�SD as the reviewer suggested.

3. Please list the total and subgroup numbers of patients with FIB-4 >6 and FIB-4 between 3.25 and 6; please list detail numbers and percentage of the parameters for patients with a FIB-4 3.5-6.

Answer: The total and subgroup numbers of patients with FIB-4 >6 and FIB-4 3.25-6 are added in Table 1. The detailed numbers and percentage of the parameters for patients with FIB-4 of 3.25–6 are listed in the footnote of Table 1.

4. The difference of each AE between the 8- and 12-week GP therapy should be compared with P-value. (Table 3)

Answer: The p-value is provided in Table 3 as the reviewer suggested.

5. Any predictors of each/total AE? The predictors of clinical significant AE (such as ALT > 5 xULN) and direct hyperbilirubinemia (> 1.5 xUNL) should be investigated. (additional information is necessary)

Answer: Logistic regression is performed to investigate the predictors of clinically significant laboratory AEs. The results were listed below for your reference but I am not sure if it is necessary to put the table in the manuscript as the predictors for elevation of ALT and bilirubin seem to be of little or no clinical implication.

[minor problems]

1. It should be effectiveness "and safety" of GP for a treatment in patients with HCV and compensated cirrhosis. (line 12 & 14, page 4)

Answer: These statements have been amended as the reviewer suggested.

2. What dose superscript "6" mean? (line 3, page 5)

Answer: Thank you for kindly reminder. The superscript “6” is deleted.

3. Please give references to support "...this AE (pruritis) can increase with longer treatment duration (line 7, page12)."

Answer: There is no any literature reporting an increased rate of pruritus with longer treatment duration by GLE/PIB. After statistical analysis, there is no significant difference in the rates of pruritus between the two groups and the statement regarding increased rate of pruritus with longer treatment duration by GLE/PIB is removed.

4. Indirect hyperbilirubinemia is less clinically significant. Please list total bilirubin > 1.5 x or 3 x ULN AND direct hyperbilirubinemia. (Table 3)

Answer: Rate of direct hyperbilirubinemia is listed in Table 3 as the reviewer suggested.

5. Footnotes of Table 1 and 3 were missed.

Answer: Footnotes of Table 1 and 3 are added.

6. The dataset in the Supporting Information was Chinese records.

Answer: The Chinese characters in the dataset are amended and the birthdates are deleted to avoid release of personal identity.

Reviewer #2: This study aimed to compare the treatment efficacy and safety between 8-week and 12-week GLE/PIB therapy for patients with compensated HCV-related liver cirrhosis. The authors showed that both 8-week and 12-week GLE/PIB therapy provide excellent antiviral efficacy and safety profiles. Although it is interesting and has clinical implications, several concerns need to be clarified.

1. In this study, 7 patients had HBV and HCV dual infections. Were they treated with antiviral therapy for HBV? How many patients developed HBV reactivation during GIE/PIB therapy for HCV infection?

Answer: There were 13 patients with dual HBV and HCV infections, seven in the 8-week group and six in the 12-week group. Among them, 4 patients were undergoing antiviral therapy for HBV, one in the 8-week group (treated with entecavir) and three in the 12-week group (two with entecavir and one with tenofovir alafenamide). No HBV reactivation was observed during GIE/PIB therapy for these 13 patients. These descriptions are added in the “Discussion”.(P15L23-P16L5)

2. Moreover, 5 patients had a history of HCC. Were they active or non-active during GLE/PIB therapy? Did they undergo therapy for HCC during DAA therapy?

Answer: Four of the 5 patients in the 8-week group had active hepatocellular carcinoma (HCC), one received transcatheter arterial chemoembolization and three received radiofrequency ablation during GLE/PIB therapy. Four of the 16 patients in the 8-week group had active HCC, none received treatment for HCC during GLE/PIB therapy. These descriptions are added in the footnote of Table 1.

3. How many patients had clinically significant portal hypertension (CSPH)? Were the rates of CSPH different between these two groups of patients?

Answer: Since this study is a retrospective analysis, there were no reliable measurements fitting the definition of clinically significant portal hypertension (CSPH). However, the number of patients with CSPH is expected to be extremely low as only patients with compensated cirrhosis were included.

4. Were the rates of biochemical response or ALT normalization different between these two groups of patients?

Answer: There was no difference regarding the rates of biochemical response and the rates and comparisons between the two groups are added in Table 2.

5. In this study, patients in the 12-week treatment group had a lower baseline eGFR than those in the 8-week treatment group. Were the dynamic changes of eGFR during and after DAA therapy different between these two groups of patients?

Answer: The 12-week treatment group had lower dynamic changes of eGFR compared to the 8-week treatment group (p = 0.0003). This result was expressed as Figure 3 and addressed in the “Discussion”.(P13L11-13)

6. It is suggested to provide a table to compare the laboratory data (such as albumin, bilirubin, ALT, platelet count, FIB-4, eGFR, etc.) at the time of end of follow-up (SVR12) between 8-week and 12-week treatment groups.

Answer: These comparisons are provided as below for your reference. The laboratory parameters with p < 0.05 at SVR12 between the two groups were the same as those at baseline, including hemoglobin, platelet count, albumin, creatinine and eGFR as shown in Table 1.

SVR12 lab data 8 Weeks

(N=91) 12 Weeks

(N=68) p value

FIB-4, median (IQR) 3.6 (2.8-4.2) 3.6 (2.9-4.6) 0.35

White blood cell count, 109 cells/L, median (IQR) 5.4 (4.5-6.6) 5.2 (4.3-7) 0.7

Hemoglobin, g/dL, median (IQR) 13.2 (12-14.4) 12.4 (11.1-14) 0.03*

Platelet count, 109 cells/L, median (IQR) 131 (107.5-151.5) 112 (93-139) 0.02*

Albumin, g/dL, median (IQR) 4.2 (4.1-4.4) 4 (3.8-4.3) 0.001*

Total bilirubin, mg/dL, median (IQR) 0.8 (0.6-1.1) 0.8 (0.7-1.1) 0.54

AST, U/L, median (IQR) 27 (21-36.5) 27 (21-33) 0.57

ALT, U/L, median (IQR) 20 (14-28.5) 20 (16-31) 0.62

Creatinine, mg/dL, median (IQR) 0.9 (0.8-1.1) 1 (0.8-2.1) 0.007*

eGFR, mL/min/1.73m2, median (IQR) 74.4 (64.2-86.1) 64 (33.1-84.5) 0.02*

Alpha-fetoprotein, ng/mL, median (IQR) 3.6 (2.4-5.4) 3.5 (2.6-5.6) 0.78

7. In Table 1, it is suggested to provide the exact p value instead of p< 0.05.

Answer: The exact p values are provided in Table 1.

---

## [Decision Letter · Decision Letter 1]

18 Jul 2022

PONE-D-22-15455R1Comparison of 8- versus 12-weeks of glecaprevir/pibrentasvir for Taiwanese patients with hepatitis C and compensated cirrhosis in a real-world settingPLOS ONE

Dear Dr. Chang,

Thank you for submitting your manuscript to PLOS ONE. After careful consideration, we feel that it has merit but does not fully meet PLOS ONE’s publication criteria as it currently stands. Therefore, we invite you to submit a revised version of the manuscript that addresses the points raised during the review process.

We look forward to receiving your revised manuscript.

Kind regards,

Chen-Hua Liu

Academic Editor

PLOS ONE

Journal Requirements:

Reviewers' comments:

Reviewer's Responses to Questions

**Comments to the Author**

1. If the authors have adequately addressed your comments raised in a previous round of review and you feel that this manuscript is now acceptable for publication, you may indicate that here to bypass the “Comments to the Author” section, enter your conflict of interest statement in the “Confidential to Editor” section, and submit your "Accept" recommendation.

Reviewer #1: All comments have been addressed

Reviewer #2: All comments have been addressed

2. Is the manuscript technically sound, and do the data support the conclusions?

Reviewer #1: Yes

Reviewer #2: Yes

3. Has the statistical analysis been performed appropriately and rigorously? 

Reviewer #1: Yes

Reviewer #2: Yes

4. Have the authors made all data underlying the findings in their manuscript fully available?

Reviewer #1: Yes

Reviewer #2: Yes

5. Is the manuscript presented in an intelligible fashion and written in standard English?

Reviewer #1: Yes

Reviewer #2: Yes

6. Review Comments to the Author

Reviewer #1: Most responses were appropriate. However, some minor problems should be modified because of newly added figure 3.

1. Please add description about dynamic changes of eGFR in the Result section. The values of eGFR at different time points cannot be identified in the figure, and these values should be mentioned in the Result section.

2. Is any difference of the dynamic changes of eGFR values or percentages (baseline-EOT; baseline-12 weeks after EOT) between 8-week and 12-week group?

Reviewer #2: (No Response)

7. PLOS authors have the option to publish the peer review history of their article (what does this mean?). If published, this will include your full peer review and any attached files.

Reviewer #1: No

Reviewer #2: **Yes: **Chien-Wei Su

---

## [Author Response · Author response to Decision Letter 1]

20 Jul 2022

Reviewer #1: Most responses were appropriate. However, some minor problems should be modified because of newly added figure 3.

1. Please add description about dynamic changes of eGFR in the Result section. The values of eGFR at different time points cannot be identified in the figure, and these values should be mentioned in the Result section.

Answer: The dynamic changes and the values of eGFR at different time points are added in the Result section as well as in the figure 3.

2. Is any difference of the dynamic changes of eGFR values or percentages (baseline-EOT; baseline-12 weeks after EOT) between 8-week and 12-week group?

Answer: The 12-week group had a lower dynamic change of eGFR compared to that of the 8-week group (p = 0.002 for the changes from baseline to EOT and p = 0.0003 for the changes from baseline to 12 weeks after EOT between the two groups).

---

## [Editor Report · Decision Letter 2]

22 Jul 2022

Comparison of 8- versus 12-weeks of glecaprevir/pibrentasvir for Taiwanese patients with hepatitis C and compensated cirrhosis in a real-world setting

PONE-D-22-15455R2

Dear Dr. Chang,

We’re pleased to inform you that your manuscript has been judged scientifically suitable for publication and will be formally accepted for publication once it meets all outstanding technical requirements.

Kind regards,

Chen-Hua Liu

Academic Editor

PLOS ONE

---

## [Editor Report · Acceptance letter]

8 Aug 2022

PONE-D-22-15455R2 

Comparison of 8- versus 12-weeks of glecaprevir/pibrentasvir for Taiwanese patients with hepatitis C and compensated cirrhosis in a real-world setting 

Dear Dr. Chang:

I'm pleased to inform you that your manuscript has been deemed suitable for publication in PLOS ONE. Congratulations! Your manuscript is now with our production department. 

Kind regards, 

on behalf of

Dr. Chen-Hua Liu 

Academic Editor

PLOS ONE